# Peer review of "Correlates of Healthy Aging in Geriatric HIV (CHANGE HIV)—CTN 314"

_viruses, 2023, doi:10.3390/v15020517_

Round 1

Reviewer 1 Report

Thank you for the opportunity to review the manuscript titled "Correlates of Healthy Aging in Geriatric HIV". Authors have reported on the sociodemographic and health characteristics of 353 participants aged 65-85 using a sex/gender stratified analysis. The results provided significant insight into the vulnerabilities and challenges faced by this special population.

Overall, the study is well organized and is expected to have a significant contribution to our knowledge on the psychosocial health of people aging with HIV. However, there are several important sections missing. Also, there is an over-emphasis on simple descriptive analyses despite data are available for more advanced work.

Specific comments to be considered:

1-   Contextual information is needed concerning the current profile of AIDS/HIV and related health services in Canada. This can help researchers compare and contrast results with other populations.

2-   Theoretical framework explaining variables selection is needed.

3-   More methodological details are needed. How were potential participants approached, identified, and invited?. Any incentives? Was it a cold calling process?

4-   Information about the survey itself are very much needed. The supplementary tables need to describe each measure (i.e., number of items, how is each item measured, examples on the items, how long it takes on average). Also detailed psychometrics are needed from both previous and current study.

5-   The survey seems to be time-consuming. How and where did participants fill it? What about participants who could not read or write or those who could not provide informed consent?

6-   Were comorbidities collected based solely on personal reports? Any confirmation? 

Thank you!

Author Response

Response to Reviewer 1 Comments

Point 1:  Contextual information is needed concerning the current profile of AIDS/HIV and related health services in Canada. This can help researchers compare and contrast results with other populations.

Response 1: Thank you for this helpful suggestion. We added this information to the second paragraph in the Introduction section, as following:

“In Canada, the health care system is publicly funded, where universal access to care includes prescription drug coverage for individuals aged 65 years and older, including antiretroviral therapy. Most people living with HIV are receiving care from primary care physicians with expertise in HIV care or specialty HIV clinics. However, there are no geriatric models of care for people living with HIV in Canada across hospital, clinic and LTC settings to address the complex social and health care needs of this population.”

Point 2: Theoretical framework explaining variables selection is needed.

Response 2: The theoretical framework is based on the WHO conceptual framework for Healthy Aging (reference 5) with the addition of HIV specific factors, as is described in the third paragraph of the Introduction section.

Point 3: More methodological details are needed. How were potential participants approached, identified, and invited?. Any incentives? Was it a cold calling process?

Response 3: Potential participants were approached by research coordinators during routine clinical visits based on their documented age, with additional details now inlcluded in the Methods subsection 2.2. Source population, eligibility, and recruitment. Details regarding compensation for participant’s time was also added:
“Potentially qualifying individuals are identified according to age with consecutive patients approached during routine clinic visits by the study site coordinator for participation and to obtain written informed consent. Participants receive compensation to cover the cost of transportation or parking at a rate of 50 CAD per study visit.”

Point 4: Information about the survey itself are very much needed. The supplementary tables need to describe each measure (i.e., number of items, how is each item measured, examples on the items, how long it takes on average). Also detailed psychometrics are needed from both previous and current study.

Response 4: All questionnaires, scales and indices used in the CHANGE HIV study have been previously described, including the number of items, how items are measures, what the items are and their psychometric properties (where available). We believe it is too onerous to present these details for each of the 37 measurement tools used in the study and instead provide relevant references for all the components included in the supplementary tables. Details regarding how long it takes to complete study visits are now included in the first paragraph of the Methods subsection 2.3 Data collection:

“On average, each study visit takes between 15 to 75 minutes to complete.”

Point 5: The survey seems to be time-consuming. How and where did participants fill it? What about participants who could not read or write or those who could not provide informed consent?

Response 5: Details regarding survey completion are now included in the first paragraph of the Methods subsection 2.3 Data collection:

“Each set of evaluations is administered over 3 visits to reduce cognitive fatigue and allow research coordinators to build rapport with study participants by the time more sensitive topics are evaluated (e.g. sexual satisfaction). On average, each study visit takes between 15 to 75 minutes to complete. Data collection is standardized across sites, utilizing standardized operational procedures developed by the project manager, who trains the site staff in the administration of the tools. During the first 2 years of the study, the majority of assessments were coordinated with routine in-person clinical visits and completed with the research coordinator on site. However, since the start of the COVID-19 pandemic, where possible, assessments were conducted virtually with participants completing questionnaires online, or over the phone or using video with a research coordinator.”

With regards to individuals who could not provide informed consent, including those who could not communicate sufficiently in English or French to provide informed consent and complete study procedures, they were excluded from the study as outlined in the Methods subsection 2.2. Source population, eligibility, and recruitment.

Point 6: Were comorbidities collected based solely on personal reports? Any confirmation? 

Response 6: Comorbidities were collected based on personal reports and confirmed through thorough review of the medical record. This is now specified in the second paragraph of the Methods subsection 2.3 Data collection:

“All components of medical history are corroborated through review of the medical record, in addition to personal report.”

Reviewer 2 Report

This study -Correlates of Healthy Aging in HIV (CHANGE HIV) is dealing with an important topic and has great potential to yield important data. It is currently in an early stage and the results are not adding much to existing knowledge. As this cohort is at an early stage of development there are shortcomings: a)There are very few women: 323 men vs 30 women. Thus the comparison of women and men for various data points is not convincing. b) The authors describe the cohort as consisting of those who aged with HIV and those who became infected in old age. How these groups differ in the selected criteria would be very interesting, but this is not even discussed. Conceivably they differ in duration of infection and ART therapies which could impact comorbidities and other Aging associated health conditions. c) There is no discussion about the ART used by this cohort. Improvements in ART over the years undoubtedly have an impact on clinical progression. d) There is no control group for comparison. How do we know how findings in people aging with HIV differ from aging without HIV. The impact of HIV on aging is not evident in this manuscript. The text in Fig 1, the conceptual model is in a small font and difficult to read.

Author Response

Response to Reviewer 2 Comments

Point 1:  There are very few women: 323 men vs 30 women. Thus the comparison of women and men for various data points is not convincing.

Response 1: Thank you for raising this point. Our intention was to highlight the patterns among men and women rather than imply significance based on the observed differences. In keeping with this, we modified the language throughout the manuscript to avoid explicit use of the word “comparing” when it comes to men and women in the sample.

Point 2:  The authors describe the cohort as consisting of those who aged with HIV and those who became infected in old age. How these groups differ in the selected criteria would be very interesting, but this is not even discussed. Conceivably they differ in duration of infection and ART therapies which could impact comorbidities and other Aging associated health conditions.

Response 2: We completely agree with the reviewer and are planning on exploring this topic in a future publication. Here we present the duration of HIV infection among participants at the time of study enrolment in 10-year increments to provide an overview of how many are long-term survivors vs infected more recently.

Point 3:  There is no discussion about the ART used by this cohort. Improvements in ART over the years undoubtedly have an impact on clinical progression. 

Response 3: Similar to point 2, addressing the impact of specific ART regimens on aging, with the majority of participants having been exposed to multiple regimens since the time of their diagnosis, was felt to be outside the scope of this manuscript. This data is being collected and will be presented in future publications.

Point 4:  There is no control group for comparison. How do we know how findings in people aging with HIV differ from aging without HIV. The impact of HIV on aging is not evident in this manuscript.

Response 4: Thank you for raising this point as this was an important consideration at the time of cohort conception. Ultimately, a control group was not included in the study because the purpose of the cohort is not to compare people living with HIV with the general population or those living with other chronic conditions. Rather, the goal is to characterize the multidimensional health status of people living with HIV age 65 and older in Canada and identify factors influencing healthy aging in this population to inform future interventions to achieve, maintain or enhance health.

Point 5:  The text in Fig 1, the conceptual model is in a small font and difficult to read. 

Response 5: An updated, larger, version of the model has been included in the manuscript.

Round 2

Reviewer 1 Report

The revised paper “Correlates of Healthy Aging in Geriatric HIV (CHANGE HIV) – CTN 314” seems to be appropriate for publishing in its current form. The authors have addressed all my comments, given detailed answers in the response letter, as well as introduced necessary additional information in the manuscript. I believe the topic of the paper will attract the attention of the journal’s readers. 

Author Response

We are very grateful to the reviewer for evaluating our revised manuscript!